# Targeted metabolomics reveals the association between central carbon metabolism and pulmonary nodules

**Yue-yang Wu[1‡], Wen-bin Shen[1‡], Jian-wei Li[1], Meng-yu Liu[1], Wen-lei Hu[1], Sheng Wang[2], Jian-jun Liu[1], Fen Huang****[1‡]\*, Qi-rong Qin[3‡]\***

**1** Department of Epidemiology and Biostatistics, School of Public Health, Anhui Medical University, Hefei, Anhui, China, **2** The Center for Scientific Research of Anhui Medical University, Hefei, Anhui, China, **3** Maanshan Center for Disease Control and Provention, Maanshan, Anhui, China

‡ YW and WS contributed equally to this work as co-first authors. FH and QQ also contributed equally to this work as co-corresponding authors.
\* fenh@ahmu.edu.cn (FH); qqr2022@163.com (QQ)

## Abstract

With the widespread application of low-dose computed tomography (LDCT) technology, pulmonary nodules have aroused more attention. Significant alteration in plasma metabolite levels, mainly amino acid and lipid, have been observed in patients of PNs. However, evidence on the association between central carbon metabolism and PNs are largely unknown. The aim of this study was to investigate the underlying association of PNs and plasma central carbon metabolites. We measured the levels of 16 plasma central carbon metabolites in 1954 participants who gained LDCT screening in MALSC cohort. The inverse probability weighting (IPW) technique was used to control for bias due to self-selection for LDCT in the assessed high-risk population. The least absolute shrinkage and selection operator (LASSO) penalized regression was used to deal with the problem of multicollinearity among metabolites and the combined association of central carbon metabolites with PNs was estimated by using quantile g-computation (QgC) models. A quartile increase in 3-hydroxybutyric acid, gluconic acid, succinic acid and hippuric acid was positively associated with the PNs risk, whereas a quartile increase in 2-oxadipic acid and fumaric acid was negatively associated with the risk of PNs in multiple-metabolite models. A positive but insignificant joint associations of the mixture of 16 metabolites with PNs was observed by using QgC models analyses. Further studies are warranted to clarify the association between circulating metabolites and PNs and the biological mechanisms.

## 1. Introduction

Lung cancer ranks first in the death spectrum of malignancies, with 18.0% mortality worldwide [1] and 28.1% nationwide of China [2]. The National Lung Cancer Screening Trial (NLST) reported that low-dose computed tomography (LDCT) screening can reduce the overall mortality rate due to lung cancer by 20% [3]. Therefore, LDCT was recommended a main

**Data Availability Statement:** All relevant data are within the paper and its Supporting Information files. All files are available from the figshare

database (http://figshare.com/) (doi:10.6084/m9.
figshare.24512401).

**Funding:** This study was supported by the project
of Special Foundation for Science and Technology
Development of Central Government Guiding
Locals (202007d07050008) and the Scientific
Research Project for Health Commission of Anhui
Province (AHWJ2021a026). The funders had no
role in study design, data collection and analysis,
decision to publish, or preparation of the
manuscript.

**Competing interests:** The author declare that they
have no known competing financial interests or
personal relationships that could have appeared to
influence that work reported in this paper.

technique for lung cancer screening in many countries [4], which enhanced the detection rate of pulmonary nodules (PNs). The evidence of both NLST [5] and a community-based LDCT lung cancer screening study in China [6] respectively found that PNs (diameter >4 mm) were estimated 27.3% and 22.9% by LDCT, indicating the detection rate of PNs is higher in the high-risk population of lung cancer.

A pulmonary nodule observed by imaging are highly likely to be diagnosed as lung cancer. Along with the increased size of the pulmonary nodule, the prevalence of lung cancer was increased ranging from 0% to 1% for nodules <6 mm, 1% to 2% for nodules 6 to 8 mm, and approximately 10% for nodules ≥8 mm [5, 7]. In addition, the detected PNs before lung cancer manifestation may cause serious anxiety and depression [8]. However, few treatment approaches available for patients with PNs other than follow-up and surgery. Therefore, it is necessary to explore the risk factors and the potential biological changes of PNs for clinical intervention. Up to now, research on the risk factors for PNs mainly focused on smoking, history of lung disease, occupational exposure and heritable factors, while the association of metabolites with PNs remained unknown. Evidence indicated that metabolites are related to the development of nodules, such as thyroid nodules and PNs [9, 10].

Central carbon metabolism, also known as energy metabolism, mainly involves glycolysis, the tricarboxylic acid cycle (TCA) and the pentose phosphate pathway (PPP). Central carbon metabolism is the primary source of energy demanded by organisms and provides precursors for other metabolisms. Previous studies showed that central carbon metabolism played an essential role in the development of pulmonary diseases. For instance, in the PPE/ LPS-induced chronic obstructive pulmonary disease(COPD) mice model [11]., the differences in urinary succinic acid, isocitric acid, and pyruvic acid were statistically significant [11]. Another bleomycin model of pulmonary fibrosis in mice found that increased level of glycolysis, TCA cycle, and PPP associated with upregulated energy production for energy demand of the fibrotic lungs [12]. A metabolomics study based on mass spectrometry showed that, compared with controls, significantly decreased levels of some plasma amino acids were observed in benign PNs [10]. Another plasma metabolomics and lipidomics study involving 1160 participants showed that metabolites associated with arginine and proline metabolism were elevated in patients with benign solitary pulmonary nodules (SPNs), while fatty acids and acylcarnitine were decreased in benign SPNs [13]. These studies suggested that amino acid and lipid metabolism are pivotal for the development of PNs. However, there are limited studies on the association between central carbon metabolism and PNs.

In summary, the present study aimed to investigate the association between central carbon metabolites and the risk of PNs in a community-based cohort of high-risk populations for lung cancer. The study to innovatively propose the association of central carbon metabolites with pulmonary nodules and explain the changes in plasma metabolites possibly caused by the occurrence of pulmonary nodules. We performed absolute quantitative Gas Chromatography-triple Quadrupole Mass Spectrometry (GC-MS/MS) metabolomics analysis of 16 central carbon metabolites mainly involved in glycometabolism and mitochondrial oxidative phosphorylation.

## 2. Materials and methods

### 2.1 Study population

The Ma'anshan Lung Screening Cohort (MALSC) is an ongoing prospective population-based cohort study and the baseline surveys were completed from June 2020 to November 2020. Volunteers aged 55–74 years were recruited and selected with achieving one of the following inclusion criteria:1) current smokers or former smokers who ceased smoking <15 years, with

intensity of smoking > = 20 pack-years 2) at least 1 year of exposure to occupational hazardous materials-such as silica, cadmium, asbestos, arsenic, beryllium, uranium, chromium, nickel, diesel exhaust, soot and ash, radon, coke oven emissions 3) history of COPD, diffuse pulmonary fibrosis or pulmonary tuberculosis 4) long history of passive smoking- more than 2 h/day and the time of duration ≥20 years at home or work 5) family history of lung cancer in the first-degree relatives 6)previous history of malignant tumor. Exclusion criteria as follows:1) has been diagnosed with lung cancer 2) hemoptysis and unexplained hemoptysis 3) clinically diagnosed as a new cancer patient within the last five years- non-pigmented skin cancer, carcinoma in situ of the cervix and localized prostate cancer were excluded 4) unexplained weight loss≥7.5kg in the past year 5) currently suffer from a serious quality of life condition. Finally, 2289 participants were recruited to perform LDCT screening. All participants were required to have lived locally for at least three years. In this study, we performed an absolute quantitative GC-MS/MS metabolomics analysis based on baseline LDCT screening results in high-risk populations. Meanwhile, we further excluded 305 participants with lung disease and finally included 1984 partcipants.

This study was approved by the Ethics Committee of Ma'anshan Center for Disease Control and Prevention (Approval No.2020001), and all participants were required to provide written informed consent for admission.

## 2.2 The definition of pulmonary nodule

The pulmonary nodule was defined as focal opacities up to 3 cm in diameter with surrounding lung parenchyma, including those abutting the pleura [14].

## 2.3 Data collection

Data on demographic and socioeconomic information (e.g., age, sex, education, income), lifestyle habits (e.g., smoking status, passive smoking, drinking, tea consumption, exposure to occupational hazards, exercise, cooking, thurification, occupational exposure to organic solvent), and personal and family history of diseases (e.g., hypertension, diabetes, lung disease, cancer, family history of cancer, family history of lung cancer in any relatives, family history of lung cancer in first-degree relatives) were collected by trained interviewers using a validated questionnaire. The level of education was divided into primary school and below, junior high school, and high school and above. Net annual household income was divided into three categories (i.e., <50000 RMB, 50000~100000 RMB, ≥100000 RMB). Smoking status was classified as never smoker, current smoker and former smoker. Current smokers were defined as those who smoke continuously or cumulatively for more than six months and smoked at least one cigarette per day. Former smokers were defined as those who had been quitting smoking for more than six months at the time of the survey. Passive smokers were considered to be those who live and work around smokers and unknowingly inhale particulate matter or various toxic substances produced by smoking. Drinkers were defined as subjects who currently drink alcohol at least once a week for more than six months. Tea drinkers were defined as subjects who currently drink tea at least once a week for more than six months. Exercisers were defined as those who exercise for more than thirty minutes at least once a week. Lung diseases include asthma, chronic bronchitis, pneumonia, emphysema, pulmonary tuberculosis, pulmonary fibrosis, silicosis, pneumoconiosis, and chronic obstructive pulmonary disease. Height, weight and blood pressure (systolic blood pressure and diastolic blood pressure) were measured by trained investigators using precise instruments. BMI (kg/m2) was calculated by dividing weight in kilograms by height in meters squared. Fasting blood samples of participants were

collected in the morning to detect blood biochemical indexes (total cholesterol, triglycerides, HDL-C, LDL-C, and fasting blood glucose) by using automated biochemical instruments.

## 2.4 Measurement of central carbon metabolites

The fasting blood samples in the early morning were collected in EDTA-K2 anticoagulant tubes, then immediately centrifuged and stored in the refrigerator at -80˚C. All plasma samples were gradient thawed on ice and then mixed in a vortex for 30 seconds. On the ice, 100 μL aliquot of each plasma sample and 20 uL [2H4]- succinic acid isotope internal standard (10 ug/mL) were precisely transferred to a 1.5 mL EP tube, then diluted with 300 uL cold methanol and vortexed for 1 min, ultrasound in an ice-water bath for 10 min. Subsequently, the EP tube containing a mixture of plasma sample and cold methanol was placed in a -20˚C refrigerator for 1 h to precipitate proteins, followed by centrifugation at 13,800 g for 15 min at 4˚C. The supernatants were collected and dried with nitrogen to obtain metabolic extracts. The dried extract samples were derived in two steps. First, 40 μL methoxyamine hydrochloride in pyridine (20 mg/mL) was added to the extract samples, and then shaken for 90 min at 37˚C with the speed of 350 rpm. Subsequently, 40 μL bis-(trimethylsilyl)–trifluoroacetamide (BSTFA) with 1% trimethylchlorosilane (TMCS) was added and incubated for another 60 min under the same conditions. After the derivatization, the mixture was centrifuged at 13800×g for 20 min. Finally, 50 uL supernatant was quantitatively transferred to sample vials for detection by gas chromatography–triple quadrupole mass spectrometry (GC-MS/MS, Agilent 7890B-7000D).

1 μL aliquot of the derivatized solution was injected into an Agilent GC system. Separation was performed on an HP-5MS fused-silica capillary column (30.0 m × 0.25 mm, 0.25 μm, Agilent).

The GC injector temperature was set to 250˚C and the injection volume was 1 μL (split ratio of 10:1). Helium was used as the carrier gas with a flow rate of 1.2 ml/min. The initial oven temperature of the GC system was set at 60˚C for 1 min. The temperature was then increased up to 100˚C at 30˚C/min, followed by a 20˚C/min increase to 220˚C, and finally increased to 280˚C at 50˚C/min. The final temperature was held for 4 minutes. The mass spectrometer (MS) was operated in electron ionization (EI) mode (70 eV, ion source temperature: 260˚C, quadrupole temperature:150˚C, transfer line temperature: 260˚C).

We consulted relevant literature, considering that there were few metabolomics studies on pulmonary nodules so far, and we also referred to the literature about the association between metabolites and lung cancer. Considering the availability of the assay, 16 candidate metabolites were finally selected for targeted metabolomics experiments to explore the association between the candidate metabolites and pulmonary nodules. The candidate metabolites and their most characteristic Q1/Q3 mass ion in GC-MS/MS were summarized in S1 Table. Missing values for each candidate were imputed by corresponding detection (LOD) were substituted by LOD/$\sqrt{2}$. LODs for 16 candidate metabolites were 7.81 ng/mL- 125.00 ng/mL. The detection rate of all metabolites is more than 85%. One QC and blank sample were run after every 20 study samples.

## 2.5 Statistical analyses

The categorical variables were expressed as numbers (percentiles) and the continuous variables were expressed as mean ± standard deviations (SDs) or median with interquartile range (IQR). The student's t-test and Mann-Whitney U test were applied to compare continuous data based on the normality of the data distribution, and the Chi-square test was applied to make comparisons of categorical data. The distributions of plasma concentrations of metabolites were

expressed as medians with IQR, and Mann-Whitney U test was performed to evaluate differences in metabolite concentrations between PNs group and non-PNs group. The Spearman's rank correlation method was used to calculate the correlations among the concentrations of the 16 metabolites.

The inverse probability weighting (IPW) technique was used to control for bias due to self-selection for LDCT in the assessed high-risk population. Briefly, two steps are required to be conducted for IPW. First, the probability of each individual being selected for the study was calculated. Next, the weight, which was the inverse of the selection probability, was calculated and included in the analysis. We assessed the balance of general characteristics between the non-screened and screened groups by calculating the standardized mean difference (SMD). SMD>0.1 was considered to be unbalanced between the two groups.

Logistic regression models with the individual weights in models were applied to estimate the weighted associations between central carbon metabolites and the risk of PNs. Plasma metabolite concentrations were used as classification variables to fit the models based on the quartiles, where the lowest quartile was the reference group. Confounders were defined as covariates on the basis of biological plausibility or statistical considerations. The odds ratios (ORs) and 95% CI were estimated based on Model 1: crude model; Model 2: adjusted for age and sex; Model 3: additionally adjusted for smoking status, drinking, exercise, occupational exposure to organic solvent, thurificatio and triglycerides.

The least absolute shrinkage and selection operator (LASSO) penalized regression was performed to deal with complex multicollinear data. In brief, the LASSO is a regression shrinkage and selection method that imposes a penalty on component regression coefficients [15]. In the LASSO regression model, with the increase of the penalty parameter lambda ($\lambda$), more penalization will be imposed on the variables, resulting in more coefficients close to zero, so it is still a sparse model [16]. Significant metabolites from the LASSO regression model were included in the multiple logistic regression simultaneously to construct a multiple-metabolite model, adjusted for potential confounding variables in Model 3.

QgC models was used to assess the joint association of relevant metabolites with the risk of PNs. QgC models is a parametric, generalized linear model-based implementation of g-computation [17], applied to estimate the joint mixture index and the importance (i.e., weights) of each component in the mixture. It provides an estimate of the association with outcome for all components in a specified mixture simultaneously increasing by one-quantile. Different from weighted quantile sum (WQS) regression, it is not limited by the assumption of directional homogeneity and also allows nonlinear and non-additive effects of the individual components of the mixture.

All statistical analyses were performed on SPSS 23.0 (version 23.0. IBM Corp., Armonk, NY, USA.) and R software (version 4.2.0, R Core Team). All statistical tests were two-sided. P < 0.05 was regarded as statistically significant.

## 2.6 Data availability

Data will be made available on request.

## 3. Results

### 3.1 Characteristics of the study population

The baseline characteristics of the high-risk populations (screened group and non-screened group) were shown in S2 Table. Compared with those who did not participate in screening, participants in screened group were more likely to be older, female, less educated, lower income, previously having smoking history or never smoking, exposed to occupational hazards, exercise regularly, cook regularly, hypertension, diabetes, to have a family history of

cancer, a family history of lung cancer in any relatives, or a family history of lung cancer in first-degree relatives (SMD>0.1). To reduce the imbalance in baseline characteristics between the screened and non-screened group, the IPW method was applied to control for selection bias. As was shown in S2 Table, the baseline characteristics were balanced between the screened and non-screened groups after IPW (all SMD<0.1). Simultaneously, we obtained the weight of each object in the screened group, so that each object in the screened group not only represented himself/herself, but also those with similar characteristics who did not participate in the screening. The weights would be used in all subsequent analyses.

The main characteristics of the screened subjects, including a comparison between those who were detected with PNs vs non-PNs, were summarized in Table 1. Out of 1954 participants, 505 males (71.6%) and 199 females (28.4%) were detected with PNs. Compared to participants free of PNs, there was a higher proportion of subjects who followed regular exercise, thurification, or occupational exposure to organic solvent in those that PNs participants.

### 3.2 Concentrations of central carbon metabolites

The plasma concentrations of alpha-ketoglutaric acid, 3-hydroxybutyric acid, gluconic acid, phosphoenolpyruvic acid, succinic acid, hippuric acid, citric acid, malic acid, L-Lactic acid, cis-aconite acid, and isocitric acid were significantly higher in PNs group than in non-PNs group ($p < 0.05$), while the plasma concentrations of 2-oxadipic acid and orotic acid were significantly lower in PNs group than in non-PNs group. Whereas with respect to other metabolites such as fumaric acid, glyceric acid, glucaric acid and maleic acid, the differences between PNs groups and non-PNs groups were not significant ($p > 0.05$) (Table 2). The Spearman correlations ranged from -0.14 to 0.71 (S1 Fig).

### 3.3 Associations of PNs with central carbon metabolites

After adjusting for potential confounders, significant associations were found for alpha-ketoglutaric acid, 2-oxadipic acid, 3-hydroxybutyric acid, phosphoenolpyruvic acid, succinic acid, hippuric acid, citric acid, malic acid, orotic acid, L-Lactic acid, cis-aconite acid and isocitric acid (p-trend < 0.05) in Model 3(Fig 1), the results in Model 1 and Model 2 were similar to those in Model 3.

Due to the strong multicollinearity of the metabolites in this study, we employed LASSO regression to filter these 16 metabolites, and selected some of them into the multi-metabolite model. Based on the LASSO regression, the multi-metabolite model included 2-oxadipic acid, 3-hydroxybutyric acid, gluconic acid, fumaric acid, succinic acid, hippuric acid, glucaric acid and orotic acid (Fig 2A and 2B). Next, a multi-metabolite logistic stepwise regression model was established to explore the mixed effects of multiple metabolites on PNs risk (Table 3). All 8 metabolites were brought into the model and additionally adjusted for the potential covariates in model 3. The association of 6 metabolites with PNs remained significant. In comparison to individuals in the lowest quartile of 3-hydroxybutyric acid, gluconic acid, succinic acid and hippuric acid, individuals in the highest quartiles showed 69%(95%CI:1.24~2.31), 70% (95% CI:1.20~2.43), 120%(95%CI:1.56~3.10) and 104%(95%CI:1.46~2.84) increased risk of PNs, respectively. Conversely, individuals in the highest quartile of two metabolites were negatively associated with the risk of PNs compared to those in the lowest quartile, with ORs of 0.28 (0.20~0.39) and 0.41 (0.28~0.58) for 2-oxadipic acid and fumaric acid, respectively.

### 3.4 QgC models analyses

Fig 3 presented the results of QgC models analyses for PNs. Overall, the 16 metabolite mixtures were positively but not significantly associated with PNs, with an OR of 1.10 (95% CI: 0.92 ~

**Table 1. Basic characteristics of PNs group and non-PNs group.**

| | non-PNs group (n = 1250) | PNs group (n = 704) | P |
|---|---|---|---|
| **Demographics** | | | |
| Age, year (mean (SD)) | 62.19 (6.84) | 62.63 (6.73) | 0.166 |
| Sex (n, %) | | | 0.609 |
| Male | 883 (70.6) | 505 (71.7) | |
| Female | 367 (29.4) | 199 (28.3) | |
| Education (n, %) | | | 0.545 |
| Primary and below | 381 (30.5) | 206 (29.3) | |
| Junior high school | 499 (39.9) | 299 (42.5) | |
| High school and above | 370 (29.6) | 199 (28.3) | |
| Income, RMB (n, %) | | | 0.808 |
| <50000 | 374(29.9) | 203 (28.8) | |
| 50000~99999 | 661 (52.9) | 383 (54.4) | |
| ≥100000 | 215 (17.2) | 118 (16.8) | |
| BMI, kg/m$^2$ (n, %) | | | 0.727 |
| <18.5 | 27 (2.2) | 11 (1.6) | |
| 18.5~23.9 | 618 (49.4) | 351 (49.9) | |
| 24.0~27.9 | 488 (39.0) | 282 (40.1) | |
| ≥28.0 | 117 (9.4) | 60 (8.5) | |
| **Lifestyle habits** | | | |
| Smoking status (n, %) | | | 0.347 |
| Never smoker | 429 (34.4) | 231 (32.8) | |
| Current smoker | 600 (48.0) | 361 (51.3) | |
| Former smoker | 221 (17.7) | 112 (15.9) | |
| Passive smoking (n, %) | | | 0.301 |
| No | 412 (33.0) | 216 (30.7) | |
| Yes | 838 (67.0) | 488 (69.3) | |
| Drinking (n, %) | | | 0.975 |
| No | 807 (64.6) | 455 (64.6) | |
| Yes | 443 (35.4) | 249 (35.4) | |
| Tea consumption (n, %) | | | 0.074 |
| No | 569 (45.5) | 291 (41.4) | |
| Yes | 681 (54.5) | 413 (58.7) | |
| Exposure to occupational hazards (n, %) | | | 0.453 |
| No | 929 (74.3) | 534 (75.9) | |
| Yes | 321 (25.7) | 170 (24.1) | |
| Exercise (n, %) | | | 0.041 |
| No | 580 (46.4) | 293 (41.6) | |
| Yes | 670 (53.6) | 411 (58.4) | |
| Cooking (n, %) | | | 0.582 |
| No | 438 (35.0) | 238 (33.8) | |
| Yes | 812 (65.0) | 466 (66.2) | |
| Thurification (n, %) | | | 0.038 |
| No | 1150 (92.0) | 628 (89.2) | |
| Yes | 100 (8.0) | 76 (10.8) | |
| Occupational exposure to organic solvent (n, %) | | | 0.008 |
| No | 1223 (97.8) | 674 (95.7) | |
| Yes | 27 (2.2) | 30 (4.3) | |

(*Continued*)

**Table 1.** (Continued)

| | non-PNs group (n = 1250) | PNs group (n = 704) | *P* |
|---|---|---|---|
| **Personal and family history of diseases** | | | |
| Hypertension (n, %) | | | 0.340 |
| No | 614 (49.1) | 330 (46.9) | |
| Yes | 636 (50.9) | 374 (53.1) | |
| Diabetes (n, %) | | | 0.208 |
| No | 1042 (83.4) | 571 (81.1) | |
| Yes | 208 (16.6) | 133 (18.9) | |
| Cancer (n, %) | | | 0.741 |
| No | 1192 (95.4) | 669 (95.0) | |
| Yes | 58 (4.6) | 35 (5.0) | |
| Family history of cancer (n, %) | | | 0.418 |
| No | 699 (55.9) | 407 (57.8) | |
| Yes | 551 (44.1) | 297 (42.2) | |
| Family history of lung cancer in any relatives (n, %) | | | 0.345 |
| No | 944 (75.5) | 545 (77.4) | |
| Yes | 306 (24.5) | 159 (22.6) | |
| Family history of lung cancer in first-degree relatives (n, %) | | | 0.316 |
| No | 959 (76.7) | 554 (78.7) | |
| Yes | 291 (23.3) | 150 (21.3) | |
| **Blood biochemical index** | | | |
| Total cholesterol, mmol/L (mean (SD)) | 4.65 (0.95) | 4.62 (0.94) | 0.493 |
| Triglycerides, mmol/L (mean (SD)) | 1.93 (1.92) | 1.77 (1.41) | 0.061 |
| HDL-C, mmol/L (mean (SD)) | 1.38 (0.35) | 1.37 (0.33) | 0.510 |
| LDL-C, mmol/L (mean (SD)) | 2.90 (0.82) | 2.93 (0.85) | 0.441 |
| Fasting blood glucose (mean (SD)) | 5.52 (1.65) | 5.62 (1.78) | 0.199 |

1.31). With the largest contribution of positive weights were hippuric acid, succinic acid and citric acid, and the largest contribution of negative weights were fumaric acid, 2-oxadipic acid and glucaric acid.

## 3.5 Stratification analysis

S3 and S5 Tables described the results of the single-metabolite model when dividing the subjects into different subgroups. A remarkably higher association between the levels of central carbon metabolites and PNs was noticed among participants of age < 65 years and males. This trend was also observed in LASSO regression (S3 and S4 Figs). In the multi-metabolite model, age has a modification effect on the association between the levels of 2-oxadipic acid, gluconic acid, fumaric acid, succinic acid, hippuric acid, malic acid and PNs (S4 Table). Additionally, sex was observed as a modifiable factor on the association between the levels of 2-oxadipic acid, 3-hydroxybutyric acid, gluconic acid, fumaric acid, succinic acid, hippuric acid and PNs (S6 Table). S6 (B) Fig shows that whole metabolite levels in females were positively associated with PNs risk (OR = 1.48, 95%CI:1.05~2.09), with the greatest positive weights for hippuric acid, phosphoenolpyruvic acid and malic acid. In addition, no significant differences were found in the association between metabolite levels and PNs in subgroups stratified by age (S5 Fig).

**Table 2. Distribution of plasma central carbon metabolites (ng/mL) in PNs group and non-PNs group.**

| Metabolites | non-PNs group (n = 1250) | PNs group (n = 704) | Z | $P^a$ |
|---|---|---|---|---|
| alpha-Ketoglutaric acid | 2191.3(1523.3~3615.8) | 2353.2(1651.2~3988.0) | -2.493 | 0.013 |
| 2-Oxadipic acid | 112.6(107.6~120.2) | 110.4(69.9~118.1) | -4.874 | <0.001 |
| 3-Hydroxybutyric acid | 8004.8(4884.0~16615.0) | 8953.4(5602.9~19794.7) | -3.741 | <0.001 |
| Gluconic acid | 958.8 (500.6~1380.9) | 1025.1(637.8~1443.0) | -2.754 | 0.006 |
| Phosphoenolpyruvic acid | 142.4 (131.8~174.3) | 150.6 (133.1~189.9) | -4.844 | <0.001 |
| Fumaric acid | 76.3(52.8~124.0) | 75.7 (51.3~123.2) | -0.234 | 0.815 |
| Glyceric acid | 837.8 (628.9~1205.5) | 873.6 (649.3~1247.1) | -1.848 | 0.065 |
| Succinic acid | 1082.3 (825.2~1440.5) | 1235.3(914.1~1671.4) | -5.909 | <0.001 |
| Hippuric acid | 443.1 (254.5~1228.4) | 774.3 (334.7~3027.2) | -8.044 | <0.001 |
| Citric acid | 15738.1 (11447.1~19739.6) | 17441.8(13818.7~22081.2) | -5.832 | <0.001 |
| Malic acid | 778.9(589.9~1100.0) | 875.6 (663.4~1198.2) | -5.134 | <0.001 |
| Glucaric acid | 64.3 (47.6~89.3) | 62.5 (43.2~92.6) | -0.533 | 0.594 |
| Orotic acid | 99.6(82.6~112.2) | 95.6(55.1~112.2) | -3.602 | <0.001 |
| L-Lactic acid | 348900.6(261142.9~668160.3) | 431015.6 (297626.8~704094.2) | -4.895 | <0.001 |
| cis-Aconite acid | 350.4(250.0~468.6) | 383.1 (268.2~552.0) | -4.599 | <0.001 |
| Isocitric acid | 3504.0(2002.9~4812.4) | 3800.0(2228.9~5323.7) | -3.536 | <0.001 |

The concentrations of metabolites were presented as median ($P_{25}$-$P_{75}$).

[a] p-Values were derived from Mann-Whitney U tests.

## 4. Discussion

Metabolomics provides comprehensive information about the functional status of cells and help to describe the phenotype of an organism. The metabolomic analysis provides a tool to reveal the metabolic pathways underlying disease. In order to explore the underlying mechanism of PNs, a targeted metabolomics study based on high-risk populations for lung cancer was performed in this study.

To the best of our knowledge, this is the first targeted metabolomics study to assess the association between central carbon metabolism and PNs in a population at high risk for lung cancer. By using a combination of mixture methods, this study has filled important knowledge gaps concerning levels of multiple plasma central carbon metabolites in the high-risk population of lung cancer and associations with PNs. Using LASSO regression with stability selection, we identified key metabolites in the mixture, and modeled the individual and joint relationships of these selected compounds with PNs. That is, a quartile increase in 3-hydroxybutyric acid, gluconic acid, succinic acid and hippuric acid was positively associated with PNs risk, whereas a quartile increase in 2-oxadipic acid and fumaric acid was negatively associated with PNs risk in multiple-metabolite models. A positive but insignificant joint association of the mixture of sixteen metabolites with PNs was observed by using QgC models analyses. These metabolites are of public health relevance owing to their partial reflection of energy metabolism, as well as the underlying role of PNs in the development of lung cancer.

In comparison of participants with PNs to controls, 3-hydroxybutyric acid was ascertained as a critical metabolic biomarker for PNs. 3-Hydroxybutyric acid, as one of the most plentiful ketone bodies in plasma, is found to perform diverse biological functions, involving epigenetic regulation, energy metabolism and oxidative stress response [18]. To date, the biological functions of 3-hydroxybutyric acid associated with PNs have yet to be reported. Neverthelss, recent studies have found that levels of hydroxybutyrate dehydrogenase are significantly increased in some non-cancer lung diseases, possibly due to its antioxidant capacity [19]. Some studies

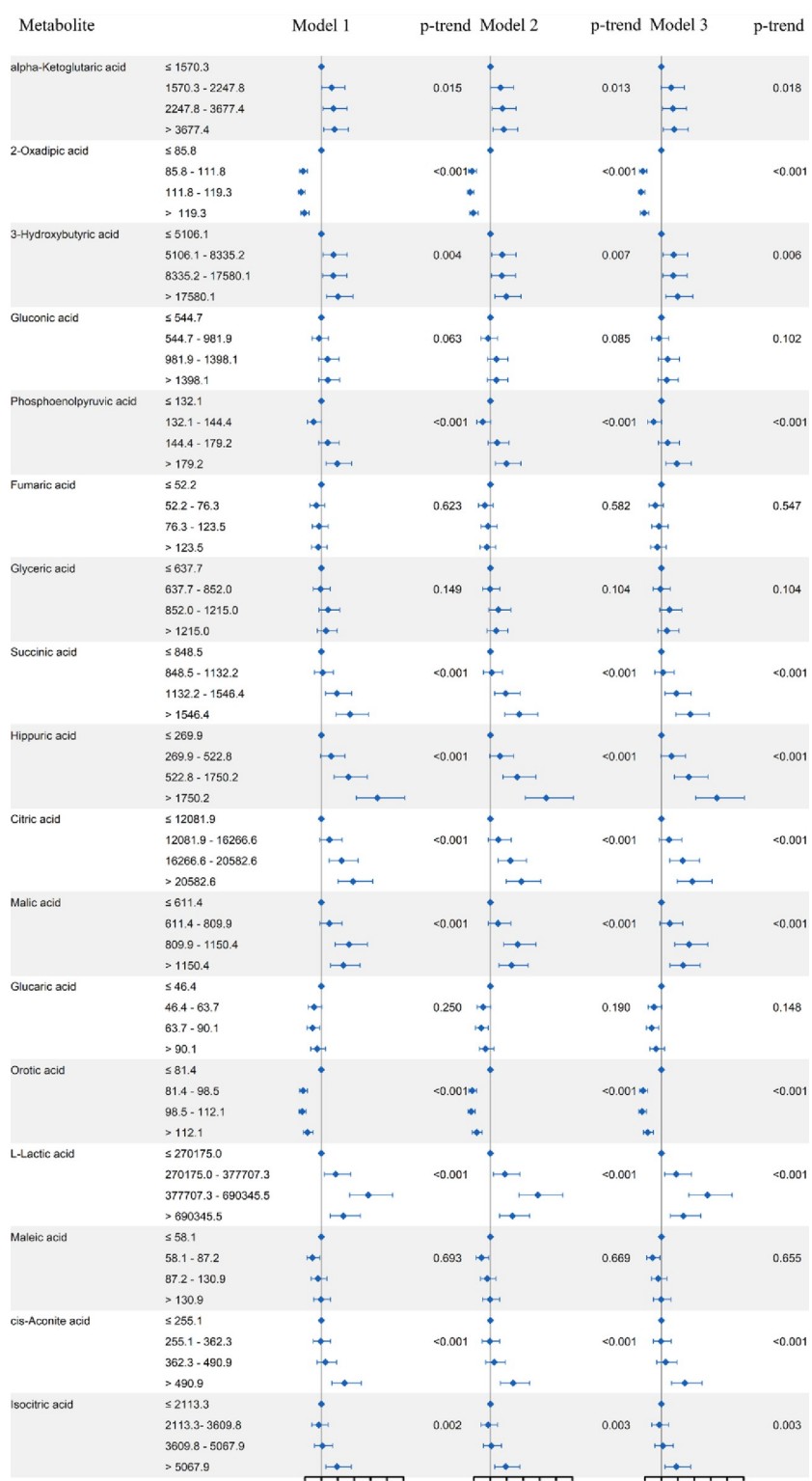

**Fig 1. Odds ratios (ORs) and 95% confidence intervals (95% CIs) according to PNs for plasma central carbon metabolites in single-metabolite model.**

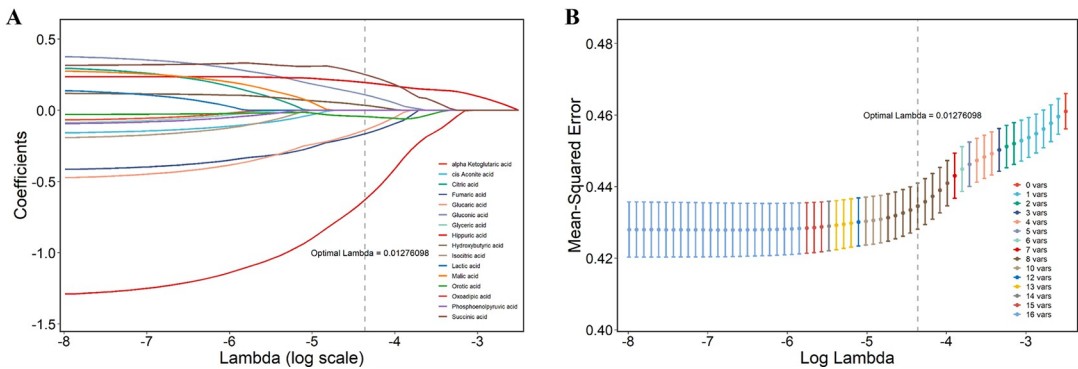

**Fig 2. The metabolites selected into the multi-metabolite model by LASSO regression.**

have demonstrated that 3-hydroxybutyrate supplementation may prevent ROS oxidation by increasing NADH oxidation [20]. In this study, high levels of 3-hydroxybutyric acid were associated with oxidative stress due to its overproduction during mitochondrial dysfunctions [21]. Further studies are warrant to reveal the exact reason for the upregulation of plasma 3-hydroxybutyric acid in participants with PNs.

The differential expression of 2-oxoadipic acid we observed has been reported in inflammatory diseases such as Behcet's disease and acne [22, 23]. 2-oxoadipic acid is a common catabolic product of the essential amino acid lysine and tryptophan, of which the tryptophan pathway is considered to play a vital role in the regulation of inflammation and immunity [24, 25]. Our findings that the levels of 2-oxoadipic acid in PNs participants were lower than in controls suggested that lysine and tryptophan catabolic pathways may be involved in the development of PNs.

Another noticeable finding is that elevated plasma gluconic acid levels were ascertained as a strong biomarker to distinguish PNs from controls. Gluconic acid is the oxidation product of glucose, its potential role and mechanism have not been elucidated. It has recently been found that gluconic acid is produced from the enzymatic reaction of glucose by regulating the activity of the enzyme glucose oxidase, in which hydrogen peroxide is released [26]. Hence, it can be inferred that gluconic acid is a marker of oxidative stress. It was reported for the first time in our study that the levels of gluconic acid were higher in PNs group than in controls. Nevertheless, the role of gluconic acid on PNs is not clear and more research needs to be conducted to validate our findings.

**Table 3. Adjusted odds ratios [95% confidence interval (CI)] for PNs according to quartiles of plasma central carbon metabolites included in the multi-metabolite model.**

| Metabolites[a] | Q1 | Q2 | Q3 | Q4 | *p*-trend | *p*-FDR |
|---|---|---|---|---|---|---|
| 2-oxadipic acid | 1.00(ref) | 0.31(0.22~0.43) | 0.25(0.17~0.35) | 0.28(0.20~0.39) | <0.001 | <0.001 |
| 3-Hydroxybutyric acid | 1.00(ref) | 1.50(1.13~1.99) | 1.50(1.12~2.00) | 1.69(1.24~2.31) | 0.004 | 0.005 |
| Gluconic acid | 1.00(ref) | 1.61(1.12~2.31) | 1.95(1.36~2.79) | 1.70(1.20~2.43) | 0.244 | 0.244 |
| Fumaric acid | 1.00(ref) | 0.80(0.60~1.06) | 0.79(0.59~1.07) | 0.41(0.28~0.58) | <0.001 | <0.001 |
| Succinic acid | 1.00(ref) | 1.19(0.89~1.59) | 1.56(1.16~2.10) | 2.20(1.56~3.10) | <0.001 | <0.001 |
| Hippuric acid | 1.00(ref) | 1.28(0.96~1.71) | 1.61(1.21~2.14) | 2.04(1.46~2.84) | <0.001 | <0.001 |

The multi-metabolite model was adjusted for age, sex, smoking status, drinking, exercise, occupational exposure to organic solvent, thurification.

[a] The concentrations of metabolites were presented in ng/mL.

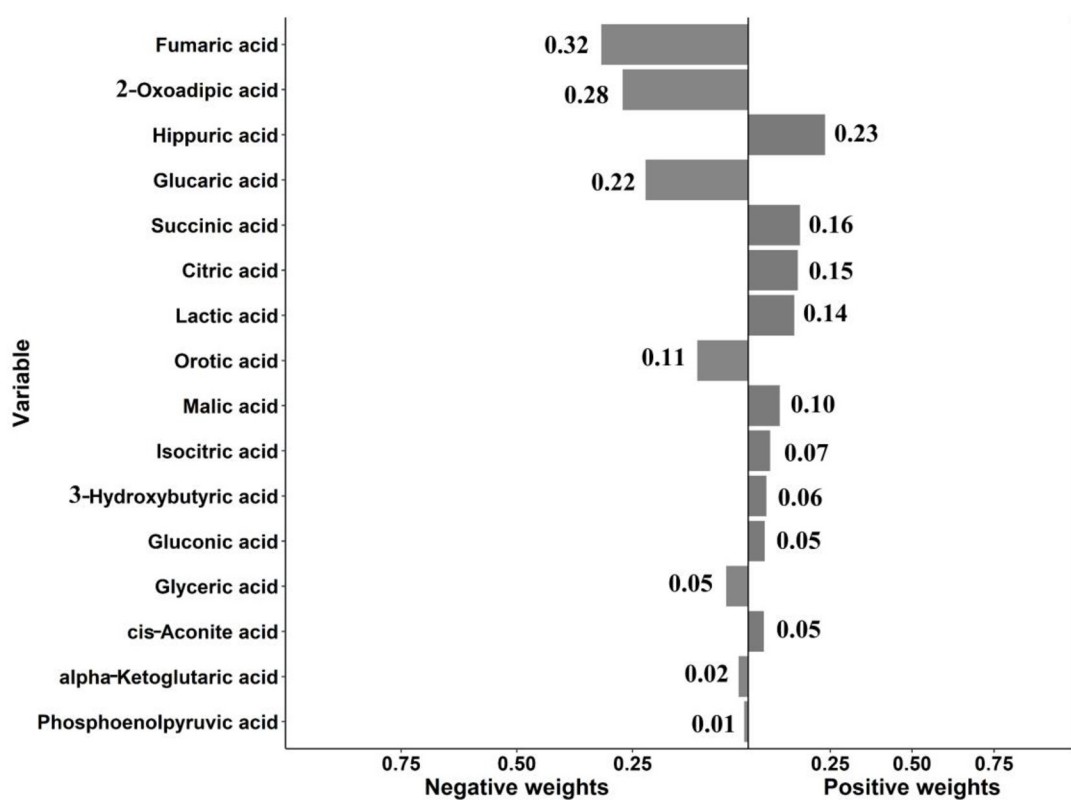

**Fig 3. Weights representing the proportion of the positive or negative partial effect for each metabolite in the quantile g-computation model with all metabolites.**

Moreover, in this study we showed that succinic acid, a tricarboxylic acid (TCA) cycle metabolite, was less abundant in healthy subjects' than in PNs patients' plasma, which is similar to the study conducted by Wu et al [27]. Another member of TCA cycle pathway, fumaric acid, also seems to play a role in PNs. Both TCA organic acids changed in plasma of participants with PNs, but in opposite trends (succinic acid elevated while fumaric acid reduced). Succinic acid is generated from succinyl-CoA by the enzymes of succinyl-CoA synthetase, and it occupies a crucial position in mitochondrial energy metabolism, serving as the only direct link between the TCA cycle and the mitochondrial respiratory chain via complex II activity [28]. Also, succinic acid has been considered to be a metabolic indicator of mitochondrial dysfunction and sepsis [29], with altered tissue concentrations under inflammation and ischemia [30, 31]. However, novel studies indicated that succinic acid released for activated macrophages can act as a pro-inflammatory local mediator [31], which could be another link between altered plasma metabolism and inflammatory responses. In contrast to our study, the levels of succinic acid were significantly reduced in urine samples from childhood allergic airway diseases [32], and in biological samples from patients with asthma COPD overlap [33]. This suggested that the expression of succinic acid in PNs may differ from other airway or lung diseases. Fumaric acid has been implicated to inhibit the degradation of HIF-1α in tumor cells so as to overcome hypoxia in several cancer types [34]. Fumaric acid can be considered a cancer metabolite in this regard. Nevertheless, the underlying rationale with regard to the opposite trend of the two metabolites in the plasma of PNs remains to be elucidated.

Hippuric acid was found to reduce the risk of PNs detection. Hippuric acid is a metabolite generated by a series of gut microbes that degrade plant (poly)phenols and aromatic amino

acids; the resulting benzoic acid is subsequently combined with glycine in the liver and kidneys and finally excreted in the urine. It was identified in a recent study as a metabolomic marker of gut microbiota diversity and found that reduced hippuric acid was associated with metabolic syndrome [35]. The gut microbiota was closely related to host health. Researchers have found that gut microbiota can also influence the development of lung disease through the gut-lung axis. However, few studies have reported the association between gut microbiota and PNs. Our study found that the levels of hippuric acid were higher in PNs group, and the dysregulation of intestinal flora may play a certain role in the development of PNs. Clearly, more detailed metabolomic studies are needed to explore the biological significance of altered hippuric acid metabolism.

We identified important metabolites associated with PNs using QgC models and found that 16 metabolites had a positive but not significant overall association with PNs. As well, a higher association between multiple metabolites and the risk of PNs was found in women. In the analysis of multi-metabolite mixtures, it is possible that the positive effect on the risk of PNs is driven by a combination of several key metabolites, involving several key metabolic pathways. Therefore, any potential intervention should target a specific set of key metabolites, rather than targeting a single metabolite or all metabolites. In subgroup analyses, compared with those age < 65 years, the positive weight of lactic acid on PNs was increased in subjects whose age ≥ 65 years, whereas the weight of malic acid was significantly decreased. Studies have shown that aging affects various metabolic pathways, many of which are associated with decreased mitochondrial function [36]. A decrease in mitochondrial function leads to a decrease in the TCA cycle, which activates the glycolytic pathway and leads to an increase in lactic acid. The positive weights of citric acid and succinic acid on PNs were remarkably higher in males than in females. It is possible that higher levels of estrogen in women protect mitochondria by increasing the expression of proteins that are part of the respiratory chain or the tricarboxylic citric acid cycle [37]. However, no statistical methods are available to elucidate the biological mechanisms, and further experimental studies are needed in the future to confirm our findings.

To the best of our knowledge, this is the first targeted metabolomics study to assess the association between central carbon metabolism and PNs in a population at high risk for lung cancer. We overcame multicollinearity among metabolites and identified the potential joint associations of multiple metabolites with PNs by using QgC models. Physical examination and experimental data were collected in strict accordance with measurement standards to reduce measurement errors as much as possible. Finally, the metabolites we identified were associated with PNs risk and might be potential targets for the therapy of PNs. More experimental studies are needed to verify our results.

It is worth noting that several limitations should be acknowledged in this study. First, due to the fact that the metabolites were detected in a single-spot plasma sample, we could not establish their stability in this study. However, the intraindividual variability and reproducibility of metabolite concentrations were estimated in some studies, and intra-class correlation coefficients were observed at moderate to high levels (range: 0.27 ~ 0.89) between samples collected during several years intervals, supporting the feasibility of employing metabolomics in epidemiological studies [38]. Second, we did not assess the impact of diet on plasma metabolites. Nevertheless, it is an overnight fast blood sample of subjects collected in the morning that we are able to minimize the interference due to the variation in intake of diet in the present study. Third, even though many of the potential confounders were adjusted for, the residual confounders could not be ruled out. Finally, the cross-sectional design limited us to infer a causal relationship between plasma metabolite levels and PNs. Long-term follow-up studies are warranted to elucidate the role of plasma metabolite levels in the development of PNs.

## 5. Conclusions

In summary, we employed a new statistical method of QgC models to evaluate the association between multiple metabolite concentrations and PNs. We found that 3-hydroxybutyric acid, gluconic acid, succinic acid and hippuric acid were positively associated with PNs, whereas 2-oxadipic acid and fumaric acid were negatively associated with PNs. A positive but insignificant joint associations of the mixture of sixteen metabolites with PNs was observed by using QgC models analyses. Further studies are warranted to investigate the association between plasma metabolites and PNs and elucidate their biological mechanisms.

## Supporting information

**S1 Fig. Spearman correlation coefficients between metabolites.**
(DOCX)

**S2 Fig. Metabolic pathway map.**
(PPTX)

**S3 Fig. The metabolites selected into the multi-metabolite model by LASSO regression in subgroups stratified by age.**
(DOCX)

**S4 Fig. The metabolites selected into the multi-metabolite model by LASSO regression in subgroups stratified by sex.**
(DOCX)

**S5 Fig. Weights representing the proportion of the positive or negative partial effect for each metabolite in the quantile g-computation model with all metabolites in subgroups stratified by age.**
(DOCX)

**S6 Fig. Weights representing the proportion of the positive or negative partial effect for each metabolite in the quantile g-computation model with all metabolites in subgroups stratified by sex.**
(DOCX)

**S1 Table. The targeted detection information of candidate metabolites by GC-MS/MS.**
(DOCX)

**S2 Table. Baseline characteristics of high-risk populations before and after inverse probability weighting.**
(DOCX)

**S3 Table. Adjusted odds ratios [95% confidence interval (CI)] for PNs in subgroups stratified by age based on the single-metabolite model.**
(DOCX)

**S4 Table. Adjusted odds ratios [95% confidence interval (CI)] for PNs in subgroups stratified by age based on the multi-metabolite model.**
(DOCX)

**S5 Table. Adjusted odds ratios [95% confidence interval (CI)] for PNs in subgroups stratified by sex based on the single-metabolite model.**
(DOCX)

**S6 Table. Adjusted odds ratios [95% confidence interval (CI)] for PNs in subgroups stratified by sex based on the multi-metabolite model.**
(DOCX)

## Author Contributions

**Conceptualization:** Yue-yang Wu, Fen Huang.

**Data curation:** Jian-wei Li, Meng-yu Liu, Wen-lei Hu, Jian-jun Liu.

**Formal analysis:** Yue-yang Wu, Wen-bin Shen.

**Funding acquisition:** Fen Huang.

**Investigation:** Yue-yang Wu, Jian-wei Li, Meng-yu Liu, Wen-lei Hu, Sheng Wang, Jian-jun Liu.

**Methodology:** Sheng Wang.

**Project administration:** Fen Huang, Qi-rong Qin.

**Writing – original draft:** Yue-yang Wu.

**Writing – review & editing:** Yue-yang Wu, Wen-bin Shen, Fen Huang, Qi-rong Qin.

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
