## [Decision Letter · Decision Letter 0]

27 Oct 2023

PONE-D-23-30022Targeted metabolomics reveals the association between central carbon.PLOS ONE

Dear Dr. huang,

Thank you for submitting your manuscript to PLOS ONE. After careful consideration, we feel that it has merit but does not fully meet PLOS ONE’s publication criteria as it currently stands. Therefore, we invite you to submit a revised version of the manuscript that addresses the points raised during the review process.

Please submit your revised manuscript by Dec 11 2023 11:59PM**.** If you will need more time than this to complete your revisions, please reply to this message or contact the journal office at plosone@plos.org. Please include the following items when submitting your revised manuscript:A rebuttal letter that responds to each point raised by the academic editor and reviewer(s). You should upload this letter as a separate file labeled 'Response to Reviewers'.A marked-up copy of your manuscript that highlights changes made to the original version. You should upload this as a separate file labeled 'Revised Manuscript with Track Changes'.An unmarked version of your revised paper without tracked changes. You should upload this as a separate file labeled 'Manuscript'.If applicable, we recommend that you deposit your laboratory protocols in protocols.io to enhance the reproducibility of your results. Protocols.io assigns your protocol its own identifier (DOI) so that it can be cited independently in the future. For instructions see: https://journals.plos.org/plosone/s/submission-guidelines#loc-laboratory-protocols. Additionally, PLOS ONE offers an option for publishing peer-reviewed Lab Protocol articles, which describe protocols hosted on protocols.io. Read more information on sharing protocols at https://plos.org/protocols?utm_medium=editorial-email&utm_source=authorletters&utm_campaign=protocols.

We look forward to receiving your revised manuscript.

Kind regards,

Bashir Sajo Mienda, PhD

Academic Editor

PLOS ONE

“This study was supported by the project of Special Foundation for Science and Technology Development of Central Government Guiding Locals (202007d07050008) and the Scientific Research Project for Health Commission of Anhui Province (AHWJ2021a026).”

4. PLOS requires an ORCID iD for the corresponding author in Editorial Manager on papers submitted after December 6th, 2016. Please ensure that you have an ORCID iD and that it is validated in Editorial Manager. To do this, go to ‘Update my Information’ (in the upper left-hand corner of the main menu), and click on the Fetch/Validate link next to the ORCID field. This will take you to the ORCID site and allow you to create a new iD or authenticate a pre-existing iD in Editorial Manager. Please see the following video for instructions on linking an ORCID iD to your Editorial Manager account: https://www.youtube.com/watch?v=_xcclfuvtxQ.

Reviewers' comments:

Reviewer's Responses to Questions

**Comments to the Author**

1. Is the manuscript technically sound, and do the data support the conclusions?

Reviewer #1: Yes

2. Has the statistical analysis been performed appropriately and rigorously? 

Reviewer #1: Yes

3. Have the authors made all data underlying the findings in their manuscript fully available?

Reviewer #1: Yes

4. Is the manuscript presented in an intelligible fashion and written in standard English?

Reviewer #1: Yes

5. Review Comments to the Author

Reviewer #1: In the manuscript entitled” Targeted metabolomics reveals the association between central carbon,” the authors investigate the association between central carbon metabolites and the risk of PNs in a community-based cohort of high-risk populations for lung cancer. They focused on a new statistical method of QgC models to evaluate the association between multiple metabolite concentrations and PNs. The question posed by the authors is interesting and data can be helpful in the clinical setting. The method has been written completely and statistical methods have been selected in an appropriate manner. However, some issues should be clarified before further consideration:

1- Abstract: The authors stated that “The aim of this study was to investigate the underlying mechanisms of PNs formation using metabolomics studies”. This statement is irrelevant to the current study as the present study has a cross-sectional design. Hence, the authors cannot provide a certain mechanistic insight for PN formation. Please rewrite it.

2- The novelty of the current study cannot be presented well. It is highly recommended that the authors provide more explanation regarding the study's novelty in the “Introduction section”.

3- In statistical analysis, authors performed logistic regression models with the individual weights in models to estimate the weighted associations between central carbon metabolites and the risk of PNs. I was wondering why the authors selected triglycerides for model 3. Please elaborate on it.

6. PLOS authors have the option to publish the peer review history of their article (what does this mean?). If published, this will include your full peer review and any attached files.

Reviewer #1: **Yes: **Solaleh Emamgholipour

---

## [Author Response · Author response to Decision Letter 0]

6 Nov 2023

Dear Editor and Reviewers, 

We are very pleased to hear from you about decision of our manuscript entitled “Targeted metabolomics reveals the association between central carbon metabolism and pulmonary nodules. (Manuscript Number: PONE-D-23-30022)”. We would like to express our sincere thanks to reviewers for thoughtful comments on the previous draft. We have carefully taken their comments into consideration in our revised manuscript, and reply to you immediately. The responses to reviewer’s comments are presented below.

Reviewer #1:

1-Abstract: The authors stated that “The aim of this study was to investigate the underlying mechanisms of PNs formation using metabolomics studies”. This statement is irrelevant to the current study as the present study has a cross-sectional design. Hence, the authors cannot provide a certain mechanistic insight for PN formation. Please rewrite it.

Answer: We have made the modifications in the Abstract.

2- The novelty of the current study cannot be presented well. It is highly recommended that the authors provide more explanation regarding the study's novelty in the “Introduction section”.

Answer: We have supplemented the novelty of this article in the “Introduction section”.

3- In statistical analysis, authors performed logistic regression models with the individual weights in models to estimate the weighted associations between central carbon metabolites and the risk of PNs. I was wondering why the authors selected triglycerides for model 3. Please elaborate on it.

Answer: It has been shown that lipid metabolism will be affected by the concentration of central carbon metabolites, and then regulate the triglyceride concentration in the body. The triglyceride is more effective after adjusting in the model three, which also verifies this point.

Thanks to your suggestions, should you need any further information, please feel free to contact us. We look forward to hearing from you.

Yours Sincerely, 

Yueyang Wu

Wenbin Shen

Fen Huang

Qirong Qin

---

## [Editor Report · Decision Letter 1]

20 Nov 2023

Targeted metabolomics reveals the association between central carbon.

PONE-D-23-30022R1

Dear Dr. HUANG,

We’re pleased to inform you that your manuscript has been judged scientifically suitable for publication and will be formally accepted for publication once it meets all outstanding technical requirements.

Kind regards,

Bashir Sajo Mienda, PhD

Academic Editor

PLOS ONE
---

## [Editor Report · Acceptance letter]

28 Nov 2023

PONE-D-23-30022R1 

Targeted metabolomics reveals the association between central carbon metabolism and pulmonary nodules. 

Dear Dr. huang:

I'm pleased to inform you that your manuscript has been deemed suitable for publication in PLOS ONE. Congratulations! Your manuscript is now with our production department. 

Kind regards, 

on behalf of

Dr. Bashir Sajo Mienda 

Academic Editor

PLOS ONE